# PAV-DiT: A Cross-modal Alignment Projected Latent Diffusion Transformer for Synchronized Audio-Video Generation

## Abstract

Sounding video generation (SVG) has emerged as a challenging task due to the inherent cross-modal temporal and semantic misalignment and the high computational costs associated with multimodal data. To address these issues, we propose the Projected Latent Audio-Video Diffusion Transformer (PAV-DiT), a novel diffusion transformer explicitly designed for synchronized audio-video synthesis. Our approach introduces a Multi-scale Dual-stream Spatio-temporal Autoencoder (MDSA) that bridges audio and video modalities through a unified cross-modal latent space. This framework compresses audio and video inputs into 2D latents, each capturing distinct aspects of the signals. To further enhance audio-visual consistency and facilitate cross-modal interaction, MDSA incorporates a multi-scale attention mechanism that enables temporal alignment across resolutions and supports fine-grained fusion between modalities. To effectively capture the fine-grained spatiotemporal dependencies inherent in SVG tasks, we introduce the Spatio-Temporal Diffusion Transformer (STDiT) as the generator of our framework. Extensive experiments demonstrate that our method achieves state-of-the-art results on standard benchmarks (Landscape and AIST++), surpassing existing approaches across all evaluation metrics while substantially accelerating training and sampling speeds. We also further explore its capabilities in open-domain SVG on AudioSet, demonstrating the generalization ability of PAV-DiT.

## 1 Introduction

Sounding Video Generation (SVG) is a multimodal content generation task that aims to synthesize dynamic videos directly from static images or textual inputs, while simultaneously generating semantically aligned and temporally synchronized audio. Its ability to produce coherent audio-video content makes SVG promising for applications in film production, virtual reality, and intelligent media. Existing SVG methods (Liu et al., 2023; Yang et al., 2025; Yariv et al., 2024; Wang et al., 2024; Zhao et al., 2025; Ruan et al., 2023; Sun et al., 2024; Xing et al., 2024) can be broadly categorized into two groups (Fig. 1a). The first is cascade generation, where video is generated first, followed by audio conditioned on the video. These methods often rely on additional alignment modules to mitigate synchronization issues (Xing et al., 2024; Zhang et al., 2024b; Comunità et al., 2024; Jeong et al., 2023; Yu et al., 2024), but typically suffer from temporal misalignment and error accumulation across cascaded stages. The second is synchronized generation, where both audio and video are generated jointly within a unified framework. While this approach reduces global alignment errors, achieving fine-grained synchronization and semantic coherence remains challenging. Representative methods (Ruan et al., 2023; Wang et al., 2024; Sun et al., 2024; Liu et al., 2023) are either trained directly in the signal space, incurring high computational costs, or adopt image DiT, which lack the capacity for fine-grained spatiotemporal modeling. Despite these efforts, fine-grained spatiotemporal alignment remains an open challenge, primarily due to the intrinsic heterogeneity between audio and video.

To systematically characterize the limitations of the SVG task, we identify three fundamental challenges that distinguish SVG from conventional unimodal video generation. **First**, the structural heterogeneity of audio and visual data introduces significant modeling difficulties. Video is a 3D tensor with dimensions for time, height, and width, necessitating complex spatiotemporal modeling.

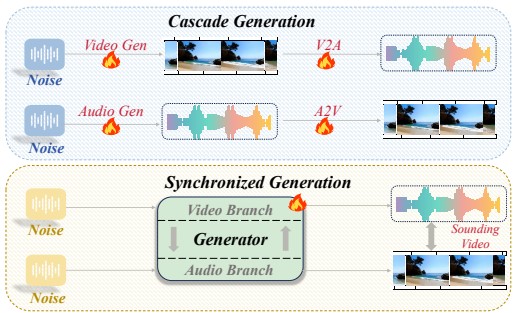 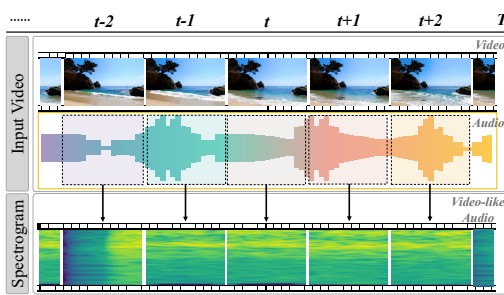

(a) Existing Audio-Video Generation methods.  (b) Video-like audio representation construction.

Figure 1: (a) Existing Audio-Video Generation methods. (b) The audio is segmented by frame, divided into audio segments in each colored square, and then converted into the Mel spectrogram sequence below. Each spectrogram has the same duration as the video frame, and the sequence is stacked along the time dimension to form a video-like audio representation ($A \in \mathbb{R}^{T \times H \times W}$), where the spectrogram acts as an image-like frame.

In contrast, audio is a 1D waveform with much higher temporal resolution (22.7 $\mu s$ vs. 33.3 $ms$ per video frame), resulting in a 1500× mismatch. This disparity complicates latent space alignment and hinders joint optimization. **Second**, achieving temporal consistency between audio and visual modalities remains highly challenging. For instance, synchronizing an explosion's sound with its corresponding visual impact requires precise multi-scale temporal modeling. However, existing methods (Ruan et al., 2023; Wang et al., 2024) often rely on shallow or global strategies, thus struggling to capture the fine-grained temporal dependencies essential for accurate synchronization. **Third**, computational inefficiency remains a critical obstacle for large-scale synchronized audio-video generation. High-dimensional inputs and multi-branch architectures incur substantial memory and computation costs, severely limiting scalability and practical deployment.

To address these challenges, we propose the Projected Latent Audio-Video Diffusion Transformer (PAV-DiT), a unified framework for efficient SVG. **First**, to mitigate the structural inconsistency between audio and video, we preprocess raw audio into video-like audio representations. As depicted in Fig. 1b, audio is segmented into frame-wise chunks, converted to mel-spectrograms, and stacked along the temporal axis. This process aligns the structure of audio with that of video, enabling them to share a compatible format and facilitating unified encoding. **Second**, we introduce the Multi-scale Dual-stream Spatio-Temporal Autoencoder (MDSA) for fine-grained spatiotemporal modeling and semantic alignment. MDSA employs orthogonal decomposition to compress both audio and video into 2D latent representations along three axes (temporal, height, and width), disentangling dynamic and static content while reducing dimensionality. It further incorporates a multi-scale attention mechanism, improving temporal coherence through Multi-scale Temporal Self-Attention (MT-SelfAttn) and aligning modality-specific features using Group Cross-Modal Attention (GCM-Attn). Finally, Bidirectional Block Cross-Attention (Bi-Block CrossAttn) enhances semantic alignment across modalities in the decoder. **Third**, to improve computational efficiency, we stack the 2D latents from MDSA into a unified 3D latent and feed it into STDiT, which efficiently models spatiotemporal dependencies via serialized spatial–temporal attention. Operating in latent space greatly reduces memory and computation, enabling high-fidelity audio–video synthesis.

We have conducted comprehensive experiments to validate the effectiveness and efficiency of our proposed method on the Landscape (Lee et al., 2022), AIST++ (Li et al., 2021) and AudioSet (Gemmeke et al., 2017) datasets. Results show that our model outperforms the state-of-the-art method and is more efficient. In particular, our model surpasses the previous SOTA model by 23.5% FVD score with 2.2× sampling speed on the Landscape dataset. Our contributions are summarized as follows:

- We propose PAV-DiT, a unified framework for synchronized audio-video generation, incorporating video-like audio representations to align audio and video in a shared latent space, improving cross-modal alignment and temporal coherence.

- We design MDSA, which introduces orthogonal feature decomposition to disentangle spatial and temporal components, reducing redundancy and enabling efficient yet expressive cross-modal fusion.

- We design a multi-scale attention mechanism, consisting of MT-SelfAttn for temporal modeling, GCM-Attn for modality-specific fusion, and Bi-Block CrossAttn for localized cross-modal integration, enhancing motion consistency and synchronization.

- Extensive experiments on AIST++ and Landscape demonstrate that PAV-DiT achieves state-of-the-art performance in generation quality and inference efficiency.

## 2 RELATED WORK

### 2.1 DIFFUSION MODELS

Diffusion Models (DMs) (Ho et al., 2020; Rombach et al., 2022) have shown strong performance in image, video, and audio generation through iterative denoising. Extending DMs to video generation (Singer et al., 2022; Ho et al., 2022b; Zhou et al., 2022; He et al., 2022; Wang et al., 2023; Ho et al., 2022a) raises challenges in spatiotemporal modeling and computational cost. VDM (Ho et al., 2022b) utilizes 3D convolutions but suffers from high computational overhead. Make-A-Video (Singer et al., 2022) and VideoLDM (He et al., 2022) decouple spatial-temporal modeling via 2D/1D convolutions and 3D-VAEs. Imagen Video (Ho et al., 2022a) and PVDM (Yu et al., 2023) reduce computational costs by applying latent compression, with PVDM using 2D latents. However, these methods are still mainly focused on unimodal video generation. Most diffusion models are based on the Transformer. Diffusion Transformers (DiTs) (Peebles & Xie, 2023; Liu et al., 2024) replace U-Net with global self-attention to better capture long-range dependencies. Latte (Ma et al., 2024), CogVideoX (Yang et al., 2024), Sora (Liu et al., 2024), and VDT (Gupta et al., 2024) improve scalability and temporal alignment through latent 3D blocks. Despite these advances, most DiTs remain modality-specific, lacking effective integration between audio and video. In contrast, our framework unifies audio and video through aligned representations and reduces complexity via hierarchical feature decoupling.

### 2.2 SOUNDING VIDEO GENERATION (SVG)

Unlike silent video generation, SVG requires the synchronous synthesis of high-quality audio-video content. Existing methods can be broadly classified into two categories. The first is cascade generation (Zhang et al., 2024a;b; Xing et al., 2024; Yang et al., 2025), while the second is synchronized generation (Liu et al., 2023; Yariv et al., 2024; Wang et al., 2024; Ruan et al., 2023; Sun et al., 2024; Zhao et al., 2025). MM-Diffusion (Ruan et al., 2023) is a pioneering method that employs diffusion models to jointly generate audio and video. It introduces two pairs of denoising diffusion models for synchronized generation and proposes a randomly shifted attention mechanism to model cross-modal consistency. MM-LDM (Sun et al., 2024) is the first latent diffusion model specifically designed for SVG, mapping audio and video to a shared semantic space via a hierarchical multi-modal autoencoder. AV-DiT (Wang et al., 2024) employs a shared pre-trained DiT backbone with lightweight adapter modules, allowing for the adaptation of a frozen image generator to audio-video tasks while reducing computational overhead. Uniform (Zhao et al., 2025) employs DiT to integrate visual and audio tokens into a unified latent space, enabling joint representation learning and audio-video generation.

Despite recent advances, existing SVG methods still face challenges such as coarse temporal alignment, limited modality correspondence, or rigid architecture designs. Some approaches rely on parameter sharing or global latent alignment (Wang et al., 2024; Sun et al., 2024), while others process raw audio or spectrograms without preserving fine-grained audiovisual consistency (Liu et al., 2023; Ruan et al., 2023; Lee et al., 2023). In contrast to prior methods, we propose PAV-DiT with hierarchical alignment for improved multi-scale temporal modeling. We further enhance cross-modal fusion by aligning modality-homogeneous latent spaces at multiple scales.

## 3 METHOD

In this section, we introduce PAV-DiT, a novel framework for synchronized audio-video generation. To tackle the challenges of cross-modal alignment and computational efficiency, PAV-DiT integrates two key components. The first is a Multi-scale Dual-stream Spatio-temporal Autoencoder

(MDSA) for encoding audio and video into a unified latent space. The second is an audio-video diffusion Transformer for generating synchronized audio-video content in this latent space. The overall framework is illustrated in Fig. 2b.

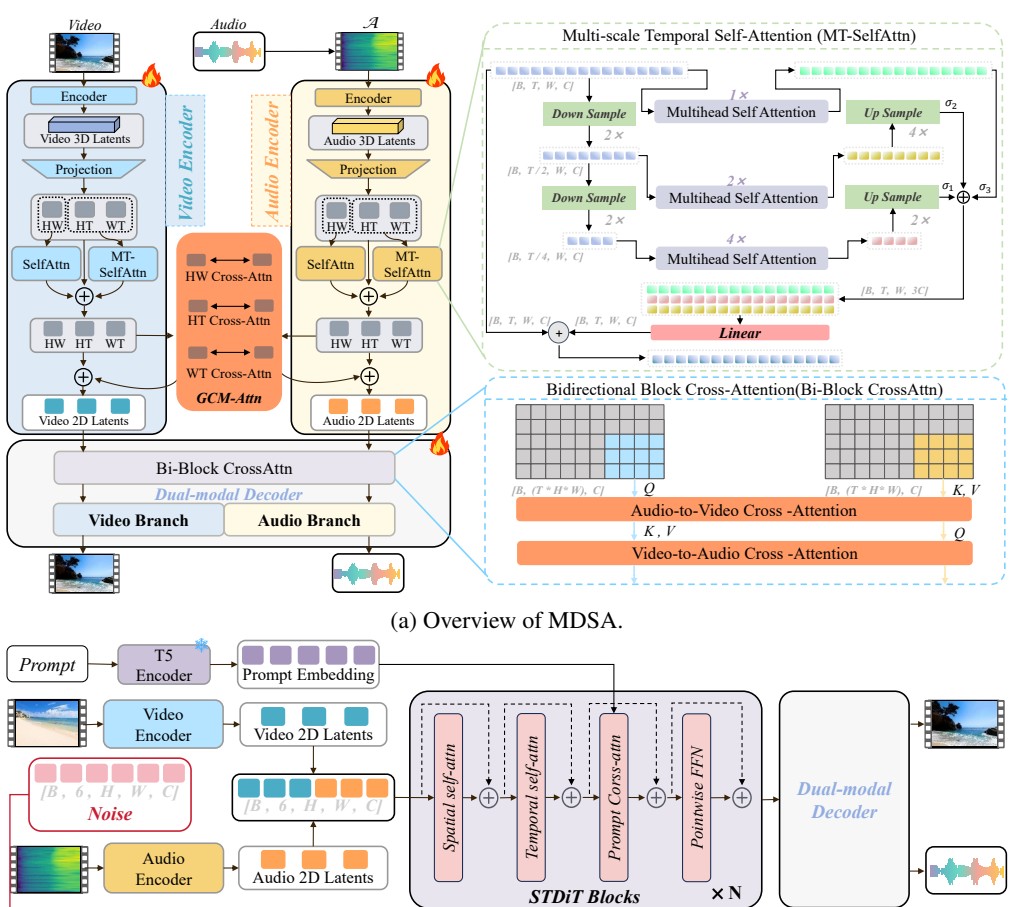

(a) Overview of MDSA.

(b) The detailed architecture of PAV-DiT.

Figure 2: (a) Given audio and video inputs, the audio is first converted into a video-like representation ($\mathcal{A}$). Both modalities are encoded via video-to-3D-latent encoders and projected into 2D latents through orthogonal decomposition. These latents are enhanced and fused using a multi-scale attention mechanism: temporal consistency (HT, WT) is modeled by MT-SelfAttn, spatial features (HW) are refined by SelfAttn, and GCM-Attn enables bidirectional cross-modal interaction. The resulting 2D latents are further processed by Bi-Block CrossAttn and decoded by a dual-modal decoder to produce synchronized audio-video outputs. (b) Audio and video latents are concatenated along the temporal axis to form a unified 3D latent representation, which serves as input to the ST-DiT. During iterative diffusion, ST-DiT progressively denoises the latents at each timestep. After the final step, the purified latents are decoded to synthesize video with temporally aligned audio-video streams.

## 3.1 MULTI-SCALE DUAL-STREAM SPATIO-TEMPORAL AUTOENCODER (MDSA)

The MDSA processes video and audio inputs to create aligned, low-dimensional latent representations. As shown in Fig. 2a, the MSDA processes both the video tensor $\mathcal{V} \in \mathbb{R}^{T \times H \times W}$ and the audio tensor $\mathcal{A} \in \mathbb{R}^{T \times H \times W}$ through three collaborative stages: dual-stream encoder, multi-scale attention mechanism, and dual-modal decoder. The encoder decomposes each modality into compact 2D latent representations, making them suitable for the subsequent diffusion process. The multi-scale attention mechanism enhances temporal coherence within each modality and facilitates cross-modal fusion, thereby improving semantic alignment and synchronization. The decoder reconstructs synchronized video and audio from the fused latent representations.

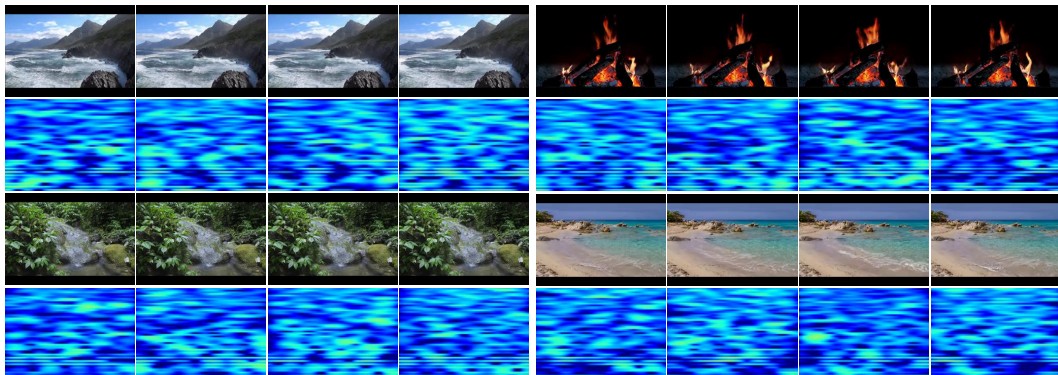

Figure 3: Results of our method on Landscape, including spectrogram visualization images and video frames.

**Input Preprocessing.** To align the structural differences between modalities, raw audio is segmented into frame-wise chunks, converted into mel-spectrograms, and stacked along the temporal axis to match the video's temporal structure. This results in a video-like audio representation $\mathcal{A} \in \mathbb{R}^{T \times H \times W}$, where each spectrogram acts as an image-like frame synchronized with the corresponding video frame, as illustrated in Fig. 1b. Unlike MM-LDM, our method supports direct reconstruction via inverse Mel transformation, avoiding reliance on neural vocoders like HiFi-GAN (Kong et al., 2020) and reducing conversion errors.

**Dual-stream Encoder.** The dual-stream encoder plays a crucial role in processing the audio and video inputs separately before they are fused. Unlike traditional methods that process video data as a 3D tensor, our approach decomposes each modality into compact 2D latent representations, enhancing efficiency and maintaining temporal coherence. Specifically, given an audio or video representation $x$, we compute a set of disentangled 2D latents $\mathbf{z} = [\mathbf{z}^t, \mathbf{z}^h, \mathbf{z}^w]$ using an encoder $f_\phi$, which consists of a video-to-3D encoder and three 3D-to-2D projectors. The encoding process can be formulated as:

$$\mathbf{u} := f_{\phi_{\text{thw}}}^{\text{thw}}(x), \text{where } \mathbf{u} \in \mathbb{R}^{T \times H' \times W'} \tag{1}$$

$$\mathbf{z} = [\mathbf{z}^t, \mathbf{z}^h, \mathbf{z}^w] \quad \text{where} \quad \begin{cases} \mathbf{z}^t = \mathbf{z}_{hw} = f_{\phi_t}^t(u) \in \mathbb{R}^{H' \times W'} \\ \mathbf{z}^h = \mathbf{z}_{tw} = f_{\phi_h}^h(u) \in \mathbb{R}^{T \times W'} \\ \mathbf{z}^w = \mathbf{z}_{wh} = f_{\phi_w}^w(u) \in \mathbb{R}^{T \times H'} \end{cases} \tag{2}$$

where $f_{\phi_{\text{thw}}}^{\text{thw}}$ is a video-to-3D-latent encoder, $f_\phi^t$, $f_\phi^h$, and $f_\phi^w$ are the 3D to 2D projection modules, $H' = H/d$ and $W' = W/d$ denote the downsampled spatial dimensions, and $T$ represents the number of temporal segments. Specifically, $\mathbf{z}_t$ encodes shared temporal information between video and audio, such as video backgrounds and audio spectral features, while $\mathbf{z}_h$ and $\mathbf{z}_w$ capture motion patterns along the height and width axes of the video, respectively. Our design is inspired by tensor decomposition and multi-view learning, where high-dimensional spatiotemporal data can often be approximated as sums of separable components:

$$\mathbf{u} \approx \sum_{r=1}^{R} \mathbf{a}_r \otimes \mathbf{b}_r \otimes \mathbf{c}_r, \tag{3}$$

with $\otimes$ denoting the outer product, and $\mathbf{a}_r \in \mathbb{R}^T, \mathbf{b}_r \in \mathbb{R}^{H'}, \mathbf{c}_r \in \mathbb{R}^{W'}$ representing variation along temporal, height, and width axes. Instead of performing full factorization, we extract the three structured projections $(\mathbf{z}^t, \mathbf{z}^h, \mathbf{z}^w)$, yielding complementary, minimally redundant latent representations. From an information-theoretic perspective, this decomposition can be interpreted as maximizing mutual information between corresponding audio-video latents along each axis while minimizing redundancy across axes, providing a principled basis for efficient, fine-grained cross-modal alignment.

**Multi-scale attention mechanism.** To ensure cross-modal consistency between audio and video, we perform temporal modeling and feature interaction on the spatiotemporal orthogonal representations $\mathbf{z}_{tw} \in \mathbb{R}^{T \times W}$ and $\mathbf{z}_{th} \in \mathbb{R}^{T \times H}$. For decoupled spatiotemporal modeling, spatial self-attention

is applied to the static content $\mathbf{z}_{hw}$, enhancing spatial feature representations and capturing long-range dependencies across different locations within each video frame. The architecture employs spatial self-attention for spatial components and introduces MT-selfAttn for temporal modeling. MT-selfAttn follows a two-step process. **First**, $2\times$ and $4\times$ average pooling is applied along the temporal axis $T$ to extract multi-scale features at three different temporal resolutions.

$$
\begin{aligned}
\mathbf{z}^{(1)} &= \mathbf{z}_t \in \mathbb{R}^{T \times S} \\
\mathbf{z}^{(2)} &= \text{AvgPool}_{2\times}(\mathbf{z}_t) \in \mathbb{R}^{T/2 \times S} \\
\mathbf{z}^{(3)} &= \text{AvgPool}_{4\times}(\mathbf{z}_t) \in \mathbb{R}^{T/4 \times S}
\end{aligned}
\tag{4}
$$

where $S$ represents the spatial dimension ($H'$ or $W'$), and $\mathbf{z}_t$ corresponds to either $\mathbf{z}_{th}$ or $\mathbf{z}_{tw}$. Self-attention is then applied to the features at each scale, with global-scale attention preserving temporal consistency, mid-scale attention capturing semantic relationships, and local-scale attention detecting transient patterns. **Second**, the attended features are upsampled back to the original temporal resolution and aggregated as follows:

$$
\tilde{\mathbf{z}}_t = \sum_{n \in \{1,2,3\}} \text{DeConv}_{k_n}\left(\text{Attn}_t(\mathbf{z}^{(n)})\right)
\tag{5}
$$

where $\text{DeConv}_{k_n}$ uses upsampling rate $k_n \in \{1, 2, 4\}$ to restore the original resolution.

The architecture hierarchically encodes visual and audio features, using decoupled spatial and temporal self-attention to capture intra-modal dependencies efficiently.

To facilitate fine-grained cross-modal fusion, we employ GCM-Attn between corresponding 2D latents from the audio and video branches. Let $\mathbf{z}_v$ and $\mathbf{z}_a$ denote video and audio latent features, respectively, derived from the three orthogonal axes $\mathbf{z}^t$, $\mathbf{z}^h$, and $\mathbf{z}^w$. For each group $\mathbf{z}$, the audio latent is updated as follows:

$$
\mathbf{z}_a = \text{CA}(\mathbf{z}_v, \mathbf{z}_a) + \mathbf{z}_a, \quad \mathbf{z} \in [\mathbf{z}^t, \mathbf{z}^h, \mathbf{z}^w]
\tag{6}
$$

Here, $\text{CA}(\cdot)$ represents a cross-attention module where the query originates from the audio latent $\mathbf{z}_a$ and the key-value pair is sourced from the corresponding video latent $\mathbf{z}_v$. The residual connection preserves modality-specific content while integrating semantically aligned video context. In a symmetric manner, the video latents are updated using the audio latents as keys and values, thereby enabling bidirectional information exchange. By combining multi-scale temporal modeling with independent spatial and temporal processing, our framework mitigates feature entanglement, eliminates redundant computation, and ensures efficient, semantically coherent cross-modal alignment.

**Dual-modal Decoding Architecture.** During decoding, the dual-modal decoder reconstructs both video and audio streams using a dual-branch architecture (Fig. 2a), enabling unified multimodal modeling. The features $\mathbf{z}'_a$ and $\mathbf{z}'_v$ are obtained by first applying the encoder for feature disentanglement, followed by a multi-scale attention mechanism to enhance the temporal coherence and semantic expressiveness of the latent representations.

$$
\mathbf{Z}_{t \times h \times w} = \mathcal{E}_t(\mathbf{z}'_{hw}) + \mathcal{E}_h(\mathbf{z}'_{tw}) + \mathcal{E}_w(\mathbf{z}'_{th}) \in \mathbf{R}^{T \times H \times W}
\tag{7}
$$

where the $\mathcal{E}$ operator expands feature maps along temporal, height, and width dimensions for alignment and fusion. Bi-Block CrossAttn is applied to model cross-modal interactions across spatiotemporal dimensions. To ensure efficiency and semantic relevance, the unified latent tensor is divided into non-overlapping blocks. Within each block, cross-attention is computed between audio and video features to capture localized correlations. The attention matrix $\mathbf{A}$ is defined as:

$$
\mathbf{A}_{i,j} = \frac{\exp(\mathbf{q}_i^\top \mathbf{k}_j / \sqrt{d})}{\sum_k \exp(\mathbf{q}_i^\top \mathbf{k}_k / \sqrt{d})},
\tag{8}
$$

where $\mathbf{q}_i$ and $\mathbf{k}_j$ are query and key vectors from the video and audio modalities, respectively. $\mathbf{A}_{i,j}$ indicates how the $i$-th video patch attends to the $j$-th audio patch, enabling block-wise semantic alignment. This design supports adaptive context modeling while maintaining efficiency through sparse attention.

Figure 4: Qualitative comparison of PAV-DiT with MM-Diffusion and MM-LDM.

Table 1: Quantitative performance comparison of multimodal video generation models on Landscape and AIST++ datasets. Results with ∗ are reproduced using released sources.

| Method | Resolution | Sampler | Landscape | | | AIST++ | | |
|---|---|---|---|---|---|---|---|---|
| | | | FVD↓ | KVD↓ | FAD↓ | FVD↓ | KVD↓ | FAD↓ |
| Single-Modal Video Generation | | | | | | | | |
| DIGAN* | $64^2$ | - | 305.4 | 19.6 | - | 119.5 | 35.8 | - |
| TATS-base* | $64^2$ | - | 600.3 | 51.5 | - | 267.2 | 41.6 | - |
| MM-Diffusion-v* | $64^2$ | dpm-solver | 237.9 | 13.9 | - | 163.1 | 28.9 | - |
| MM-Diffusion-v+SR* | $64^2$ | dpm-solver+DDIM | 225.4 | 13.3 | - | 142.9 | 24.9 | - |
| MM-LDM-v* | $64^2$ | DDIM | 122.1 | 6.4 | - | 83.1 | 13.1 | - |
| MM-Diffusion-v+SR* | $256^2$ | dpm-solver+DDIM | 347.9 | 27.8 | - | 225.1 | 51.9 | - |
| MM-LDM-v* | $256^2$ | DDIM | 156.1 | 13.0 | - | 120.9 | 26.5 | - |
| PAV-DiT-v | $256^2$ | Reflect Flow | **90.9** | **7.8** | - | **84.2** | **13.3** | - |
| Multi-Modal Generation | | | | | | | | |
| MM-Diffusion-svg+SR* | $64^2$ | dpm-solver+DDIM | 211.2 | 12.6 | 9.9 | 137.4 | 24.2 | 12.3 |
| MM-LDM-svg* | $64^2$ | DDIM | 77.4 | 3.2 | 9.1 | 55.9 | 8.2 | 10.2 |
| MM-Diffusion-svg+SR* | $256^2$ | dpm-solver+DDIM | 332.1 | 26.6 | 9.9 | 219.6 | 49.1 | 12.3 |
| MM-LDM-svg* | $256^2$ | DDIM | 105.0 | 8.3 | 9.1 | 105.0 | 27.9 | 10.2 |
| AV-DiT* | $256^2$ | - | 172.7 | 15.4 | 11.2 | 68.8 | 21.0 | 10.2 |
| PAV-DiT wo/text | $256^2$ | Reflect Flow | 87.3 | 7.7 | 8.7 | 85.6 | 19.8 | 9.8 |
| PAV-DiT | $256^2$ | Reflect Flow | **80.3** | **7.3** | **8.5** | **77.6** | **18.2** | **9.4** |
| 200 Samples (Follow the See&Hear experimental setup) | | | | | | | | |
| See&Hear* | $256^2$ | - | 326.2 | 9.2 | 12.7 | - | - | - |
| AV-DiT* | $256^2$ | - | 260.5 | 9.2 | 14.1 | - | - | - |
| PAV-DiT | $256^2$ | Reflect Flow | **240.3** | **9.0** | **12.2** | - | - | - |

## 3.2 Cross-Modal Diffusion Transformer for Synchronized Generation

As illustrated in Fig. 2b, PAV-DiT first encodes video and audio inputs into three orthogonal 2D latent representations using the MDSA encoder: $\mathbf{z}_v = [\mathbf{z}_v^t, \mathbf{z}_v^h, \mathbf{z}_v^w]$ for video and $\mathbf{z}_a = [\mathbf{z}_a^t, \mathbf{z}_a^h, \mathbf{z}_a^w]$ for audio. These representations respectively capture spatiotemporal visual features and time-frequency audio characteristics. To enhance temporal consistency in modeling the joint data distribution $p_{\text{data}}(\mathbf{z}_v, \mathbf{z}_a)$, we adopt a temporal stacking strategy within the STDiT framework, where modality-specific latent tensors are concatenated along the temporal dimension to form a unified six-frame latent sequence.

$$\mathbf{P} = \text{Stack}([\mathbf{z}_v^t, \mathbf{z}_v^h, \mathbf{z}_v^w, \mathbf{z}_a^t, \mathbf{z}_a^h, \mathbf{z}_a^w]) \in \mathbb{R}^{6 \times H \times W} \tag{9}$$

The $\mathbf{P}$ serves as input to the STDiT generator, which jointly models cross-modal interactions. The resulting latent outputs are then decoded by a dual-modal decoder to synthesize videos with temporally aligned audio and video content. The framework enables efficient multimodal fusion through two key innovations. First, we represent the audio and video latents $\mathbf{z}_v$ and $\mathbf{z}_a$ as six contiguous frames within a unified tensor $\mathbf{P}$. This stacking strategy preserves local spatiotemporal continuity and allows the use of standard video diffusion transformers without architectural modifications. Second, this unified latent representation enables joint modeling of audio-video semantics through a single spatiotemporal attention mechanism, while maintaining computational efficiency.

# 4 EXPERIMENT

## 4.1 EXPERIMENTAL SETUPS

**Datasets and Evaluation Metrics.** Following Ruan et al. (Ruan et al., 2023), we evaluate our model on two benchmark datasets: Landscape (Lee et al., 2022) AIST++ (Li et al., 2021) and AudioSet (Gemmeke et al., 2017). Both datasets are preprocessed into 16-frame video clips with each frame resized to $256^2$ resolution.Details are provided in the supplementary material. We use Fréchet Video Distance (FVD) and Kernel Video Distance (KVD) to assess video quality, and Fréchet Audio Distance (FAD) to measure audio fidelity, aligning with prior studies (Sun et al., 2024; Ruan et al., 2023; Wang et al., 2024). All videos generated by PAV-DiT are synthesized at $256^2$ resolution.

**Implementation Details.** The autoencoder and diffusion model are trained using the Adam (Kingma & Ba, 2017) and AdamW (Loshchilov & Hutter, 2019) optimizers, respectively. The dual-stream encoder is based on Timesformer (Bertasius et al., 2021), which serves as the backbone for projecting video into 3D latents. A two-stage training scheme is adopted. We first minimize perceptual loss, followed by adversarial loss and KL divergence loss ($\beta = 6\mathrm{e}^{-6}$). We adopt STDiT as the generative model, and all reported results in this paper are obtained using 16 STDiT blocks. Video and audio discriminators are jointly trained with equal weighting (0.5). A linear noise schedule and reflected-flow sampling are used to accelerate inference. Details are provided in the supplementary material.

## 4.2 QUANTITATIVE AND QUALITATIVE COMPARISON

**Qualitative Comparison.** Fig. 4 shows a qualitative comparison among PAV-DiT, MM-LDM, and MM-Diffusion. MM-Diffusion produces blurry, low-detail samples, while MM-LDM achieves clearer outputs with improved audio-video alignment but still lags behind PAV-DiT in realism and fidelity. Fig. 3 further illustrates PAV-DiT's superior generation quality on the Landscape dataset. Moreover, the autoencoder yields reconstructions visually indistinguishable from ground truth, and our audio reconstruction results (Appendix) demonstrate that recovered waveforms closely align with the originals, validating the effectiveness of our heterogeneous modality unification.

**Performance Comparison with Previous Methods.** We quantitatively compare our method with prior approaches to validate the effectiveness of PAV-DiT. As shown in Table 1, we quantitatively compare our method with previous approaches to validate the effectiveness of PAV-DiT in the Sounding Video Generation task. When conditional inputs are provided, PAV-DiT achieves average improvements of 7.3 in FVD on the Landscape dataset and 8.0 in FVD on the AIST++ dataset over unconditional generation. Furthermore, it achieves average gains of 0.3 in FAD and 1.0 in KVD across both datasets, demonstrating its capacity to capture cross-modal correlations. These improvements are attributed to the dual-stream encoder, which enhances cross-modal representation alignment and improves generation quality for both video and audio. Benefiting from an end-to-end training framework at a resolution of 256×256, in which the autoencoder directly processes native-resolution videos, PAV-DiT can generate high-quality outputs without relying on an additional super-resolution module. Specifically, PAV-DiT achieves FVD of 80.3, KVD of 7.3, and FAD of 8.5 on the Landscape dataset, and FVD of 77.6, KVD of 18.2, and FAD of 9.4 on the AIST++ datasets, establishing new state-of-the-art performance at this resolution. Notably, in contrast to conventional methods that utilize DDPM or DDIM samplers—typically requiring 100–200 steps—PAV-DiT incorporates Reflected Flow Sampling, which generates high-quality videos in only 30 steps. This approach accelerates inference by a factor of 3–6×.To evaluate the generalization and scalability of PAV-DiT, we conducted experiments on a larger open-domain dataset. Following the protocol of MM-Diffusion, we selected 100K high-quality videos from AudioSet (Gemmeke et al., 2017). As reported in Table 2, PAV-DiT consistently outperforms prior methods, demonstrating its effectiveness at larger scales.

**Efficiency Comparison.** As shown in Table 3, PAV-DiT substantially improves both training and inference efficiency. Unlike MM-Diffusion, which operates in signal space and runs out of memory at $256^2$, PAV-DiT leverages MDSA to process $256^2$ efficiently. Both MM-LDM and PAV-DiT integrate an autoencoder with the DiT generator, while MM-LDM* and PAV-DiT* isolate DiT

Table 2: Quantitative comparison with scaled-up data and model for open-domain generation.

| Model | #P | FVD↓ | KVD↓ | FAD↓ |
|---|---|---|---|---|
| MM-Diffusion | 134M | 649.8 | 34.6 | 2.9 |
| MM-LDM-S | 131M | 185.8 | 10.1 | 1.59 |
| MM-LDM-B | 384M | 181.5 | 9.5 | 1.55 |
| MM-LDM-L | 1543M | 164.1 | 8.5 | 1.52 |
| PAV-DiT | 702M | 148.7 | 8.4 | 1.51 |

Table 3: Efficiency comparison of PAV-DiT with MM-Diffusion and MM-LDM.

| Method | Res. | Train/Step | Infer/Sample |
|---|---|---|---|
| MM-Diffusion | $64^2$ | 1.70s | 33.8s |
| MM-Diffusion | $128^2$ | 2.36s | 90.0s |
| MM-LDM | $256^2$ | 0.46s | 70.0s |
| MM-LDM* | $256^2$ | 0.38s | 8.7s |
| PAV-DiT (ours) | $256^2$ | 0.44s | 17.7s |
| PAV-DiT* (ours) | $256^2$ | **0.32s** | **3.93s** |

generation performance by precomputing and storing latents, with PAV-DiT* further boosting efficiency. At a batch size of 2, PAV-DiT trains at 0.31s per step, faster than MM-LDM (0.38s) and MM-Diffusion (2.36s at $128^2$). For inference, Reflected Flow Sampling (Xie et al., 2024) reduces sampling steps from 100–200 to 30, cutting runtime to 3.9s per sample—yielding 2.2× and 22.5× speedups over MM-LDM (8.7s) and MM-Diffusion (90s), respectively.

### 4.3 HUMAN EVALUATION

We conducted a human evaluation on 1,500 samples from PAV-DiT, MM-Diffusion, and MM-LDM using the Landscape dataset, following MM-Diffusion's protocol. Two annotators rated each sample on a 5-point scale across three criteria: Audio Quality (AQ), Video Quality (VQ), and Audio-Video Alignment (A-V). As shown in Table 4, PAV-DiT consistently outperforms baselines, achieving relative gains over MM-LDM of 8.1% in AQ, 7.9% in VQ , and 9.4% in A-V.

Table 4: Human Evaluation Results

| Method | AQ↑ | VQ↑ | A-V↑ |
|---|---|---|---|
| MM-Diffusion | 2.46 | 2.10 | 2.99 |
| MM-LDM | 2.98 | 3.68 | 3.29 |
| PAV-DiT | 3.22 | 3.97 | 3.60 |

### 4.4 ABLATION STUDY

We perform an ablation study on the architecture of MDSA, with the results summarized in Table 5. The base model demonstrates strong performance (rFVD: 29.5, rKVD: 1.3, FAD: 8.5), significantly outperforming PVDM (rFVD: 70.2) and MM-LDM (rFVD: 53.9), indicating its superiority in audio-video generation quality and synchronization. To assess the impact of the multi-scale attention mechanism, we remove MT-selfAttn, GCM-Attn, or Bi-Block CrossAttn leads to higher rFVD scores of 35.1, 39.2, and 45.6, respectively, confirming the effectiveness of our attention design. Ablating KL regularization results in a degraded performance, with rFVD increasing to 60.2, demonstrating the benefit of regularization-based fine-tuning.

Table 5: Ablation study of MDSA (PAV-DiT) on the Landscape dataset

| Model | rFVD | rKVD | FAD |
|---|---|---|---|
| PVDM | 70.2 | 4.1 | 9.0 |
| MM-LDM | 53.9 | 2.4 | 8.9 |
| **MDSA (PAV-DiT)** | **29.5** | **1.3** | **8.5** |
| *multi-scale attention mechanism* | | | |
| − MT-selfAttn | 35.1 | 2.2 | 8.6 |
| − GCM-Attn | 39.2 | 2.3 | 8.7 |
| − Bi-Block CrossAttn | 45.6 | 2.4 | 8.8 |
| *Other ablations:* | | | |
| − finetune with KL loss | 60.2 | 3.2 | 8.9 |
| − adversarial loss | 134.5 | 8.3 | 9.8 |

## 5 CONCLUSION

We introduce PAV-DiT, a novel diffusion transformer designed for the SVG task. We propose a unified projection autoencoder that maps both audio and video into a shared latent space by projecting 3D data into a 2D representation. Furthermore, the use of a multi-scale dual-stream spatiotemporal autoencoder and the multi-scale attention mechanism strengthens temporal synchronization and bridges the semantic gap between modalities. Built upon the STDiT architecture, PAV-DiT enables rich cross-modal interactions during the generation process. Our method achieves new state-of-the-art results across multiple benchmarks, demonstrating superior efficiency and promising adaptability.

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

## A APPENDIX

### A.1 DATASETS

**Landscape Dataset.** The Landscape dataset focuses on high-fidelity audiovisual synchronization of natural scenes. It comprises 928 landscape videos crawled from YouTube, covering nine representative natural scenarios such as rainfall, splashing water, thunderstorms, and underwater bubbles. Each video clip is annotated with scene category, weather conditions, and types of acoustic events (e.g., "heavy rain + thunder"), supporting fine-grained conditional generation tasks. After preprocessing, the dataset yields 1,000 non-overlapping 10-second video clips, with a total duration of approximately 2.7 hours. The audio tracks feature a high dynamic range of environmental sounds that are tightly aligned with the visual scenes (e.g., thunder sounds coinciding with lightning flashes), providing naturally aligned annotations for cross-modal learning tasks.

**AIST++ Dataset.** AIST++ is constructed based on the AIST street dance database and consists of 1,020 dance video clips (with a total duration of 5.2 hours), accompanied by 60 copyright-free music tracks spanning 10 dance genres (e.g., Hip-Hop, Krump, Ballet Jazz). The dataset includes 85% basic choreography and 15% freestyle movements, enhancing the model's ability to adapt to diverse musical styles. It provides 9 camera pose parameters, 17 COCO-format 2D/3D keypoints, 24-dimensional SMPL pose parameters, and global motion trajectories. Its core value lies in the precise spatiotemporal alignment between dance movements and music. Through multi-view camera calibration and SMPL-based 3D human motion reconstruction, the dataset offers 3D motion sequences with joint rotations and displacement information, along with annotated music beat timestamps.

### A.2 IMPLEMENTATION DETAILS

**Detailed description of training objective** The proposed multi-modal adversarial training objective jointly optimizes reconstruction constraints and distribution alignment through a dynamically scheduled optimization framework. The overall loss function is defined as:

$$\mathcal{L}_{\text{total}} = \sum_{m \in \{\text{video,audio}\}} \omega_m \left( \mathcal{L}_{\text{rec}} + \mathcal{L}_{\text{perc}} + \gamma(\mathcal{L}_{\text{adv}}^{(G)} + \mathcal{L}_{\text{fm}}) \right) \tag{10}$$

where $\omega_m = 0.5$ balances cross-modal weights and $\gamma$ denotes the dynamic adversarial activation factor. Specifically, the multimodal reconstruction loss combines pixel-level fidelity with latent space regularization:

$$\mathcal{L}_{\text{rec}} = \underbrace{4.0 \cdot \|\mathbf{x} - \hat{\mathbf{x}}\|_1}_{\text{pixel fidelity}} + \underbrace{6 \times 10^{-6} \cdot D_{\text{KL}} \left( \mathcal{Q}(\hat{\mathbf{x}}) \parallel \mathcal{P}(\mathbf{x}) \right)}_{\text{distribution alignment}} \tag{11}$$

with KL-divergence computed using batchmean reduction: $\text{KL}(p \parallel q) = \sum p(x) \log \frac{p(x)}{q(x)}$.

We adopt a two-stage training strategy to stabilize optimization. During the first stage, both the adversarial loss $\mathcal{L}_{\text{adv}}^{(G)}$ and KL regularization are disabled, allowing the model to focus on basic reconstruction. In the second stage, we enable adversarial training and KL divergence to enhance visual fidelity and latent alignment. The dynamic adversarial activation is governed by:

$$\gamma = \begin{cases} 0 & \text{if } t < t_{\text{threshold}} \\ 1.0 & \text{otherwise} \end{cases} \tag{12}$$

where $t_{\text{threshold}}$ denotes the training step at which the discriminator becomes active (controlled via the `disc_start` parameter in code).

**Training Details.** For all experiments, we use a batch size of 32 and a learning rate of $1 \times 10^{-4}$ to train the autoencoders. Training continues until both FVD and PSNR metrics converge. For 3D-to-2D projection, we employ a 4-layer Transformer with 4 attention heads, a hidden dimension of 384, and an MLP dimension of 512. The latent codebook dimensionality is set to 4. For diffusion model training, we use a batch size of 64 and the same learning rate of $1 \times 10^{-4}$. Additional architectural hyperparameters, we basically followed the parameters of Opensora Liu et al. (2024), but we used 16 layers in STDiT. Specifically, we set the codebook channel $C = 4$ and the patch size to $4 \times 4 \times 1$, such that a video of size $256 \times 256 \times 16 \times 3$ is encoded into a latent vector of size $(32 \times 32 + 32 \times 16 + 32 \times 16) \times 4 = 8192$.

**Metric.** To ensure a fair comparison with prior work, we adopt consistent settings for quantitative evaluation. For Fréchet Video Distance (FVD) and Fréchet Audio Distance (FVD), we follow the fixed protocol proposed by StyleGAN-V Skorokhodov et al. (2021). Unlike the standard protocol—which first preprocesses the dataset into fixed-length video clips before computing real statistics—the StyleGAN-V protocol samples video data first, then randomly extracts fixed-length clips. This adjustment addresses bias introduced when long videos dominate the dataset, skewing the statistics due to their excessive number of clips. Following MM-Diffusion, we sample 2,048 videos (or the full dataset if it contains fewer samples) to compute the real distribution and another 2,048 videos to evaluate the generated samples.

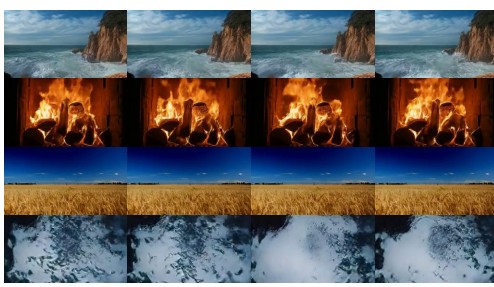

Figure 5: Video reconstruction results of our MDSA on the Landscape dataset

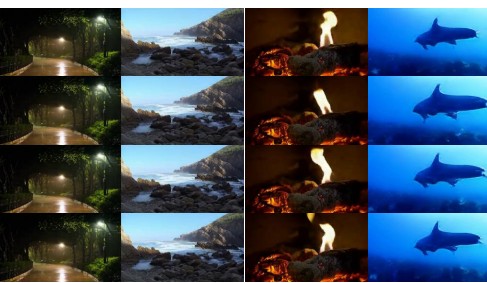

Figure 6: Video reconstruction results of our MDSA on the Landscape dataset

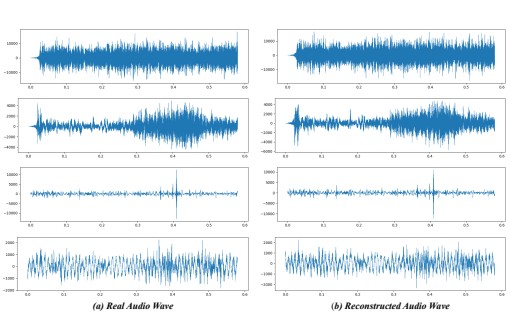

Figure 7: Audio reconstruction results of our MDSA on the Landscape dataset

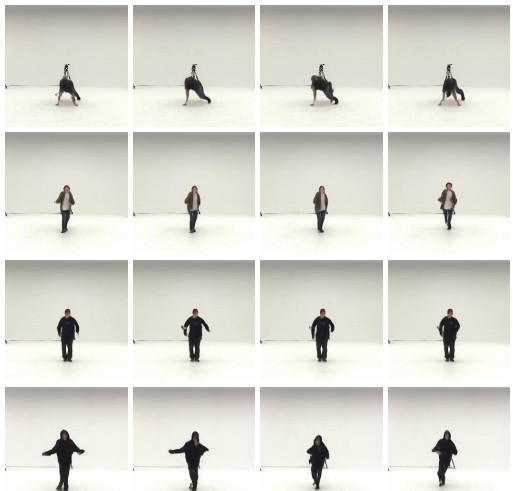

Figure 8: Results of our method on AIST++.

### A.3 QUANTITATIVE COMPARISON

**Autoencoder.** Evaluated on two datasets, it surpasses the single-stream baseline with gains of 1.40 dB PSNR and 14.6 FVD in video, and 0.85 dB PSNR and 0.5 FAD in audio—demonstrating the effectiveness of hierarchical attention-enhanced dual-branch reconstruction. The detailed results are provided in the Fig. 6.

Table 6: Comparison of our autoencoder performance with the baseline (PVDM)

| Method | FVD↓ | FAD↓ | PSNR↑ |
|---|---|---|---|
| PVDM-v | 30.3 | - | 31.34 |
| PVDM-a | - | 9.4 | 35.93 |
| PAV-DiT-v | 18.7 | - | 32.19 |
| PAV-DiT-a | - | 8.9 | 37.33 |

### A.4 QUALITATIVE RESULTS

**Autoencoder.** Fig. 5 and Fig. 6 shows the results of our MDSA reconstruction of the Landscape dataset. As can be seen, our MDSA produces high-quality synthetic results overall. The precise

reconstruction of the layered rock structure of the coastal cliffs, the instantaneous shape of the splashing waves and their gradual transition back to the sea surface, and the natural transition from the bright flame core to the orange outer flame all demonstrate the excellent performance of our autoencoder. Figure 7 illustrates the audio reconstruction performance of our method. Subfigure (a) shows the ground-truth audio waveform, while subfigure (b) depicts the reconstructed waveform obtained after encoding and decoding via our autoencoder, followed by inverse Mel-spectrogram transformation. The close similarity between the two waveforms demonstrates that our autoencoder effectively preserves high-fidelity audio content. This accurate reconstruction of both audio and video provides a solid foundation for the subsequent high-fidelity generation within the diffusion model.

**Diffusion generator.** We present qualitative results of PAV-DiT in Fig. 8 and Fig. 9, showcasing generated samples on the AIST++ and Landscape datasets, respectively. The synthesized videos exhibit high visual fidelity and realism, demonstrating the effectiveness of PAV-DiT in generating temporally coherent and semantically meaningful audiovisual content. We also present results for text-to-video generation in the Fig. 9. Since category names are used as text conditions during training, the model can generate corresponding videos when prompted with queries such as "a video of <x>", demonstrating its capacity to generalize from text-based inputs.

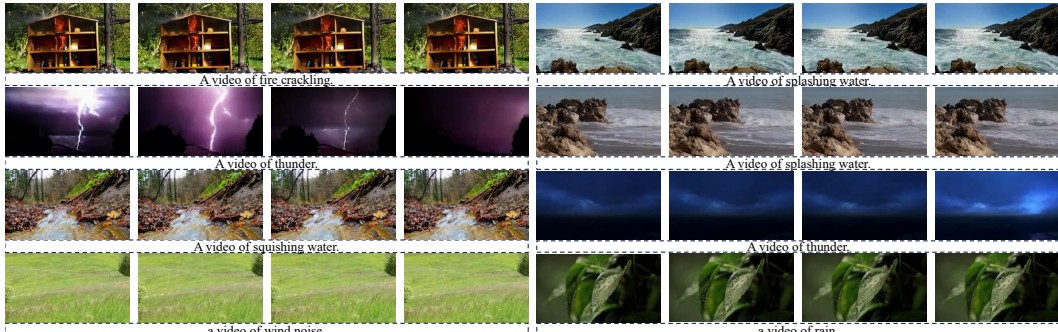

Figure 9: Our method's text-conditional guided generation results on the landscape dataset.

## B    THE USE OF LLM

Throughout the preparation of this paper, we employed a large language model (LLM) to enhance the writing and correct grammatical mistakes.

## C    REPRODUCIBILITY STATEMENT

Implementation details, evaluation protocols, and dataset descriptions are provided in the main text and appendix. Complete proofs are also included in the main text. The full source code will be released upon acceptance.

