# OpenReview forum: "PAV-DiT: A Cross-modal Alignment Projected Latent Diffusion Transformer for Synchronized Audio-Video Generation"
_ICLR.cc/2026/Conference — ICLR 2026 Conference Withdrawn Submission_

### Official Review · Reviewer_tTx8 · 2025-10-31

**Soundness:** 3
**Presentation:** 3
**Contribution:** 3
**Rating:** 4
**Confidence:** 5

**Summary:**

This paper introduces PAV-DiT, a novel framework for synchronized audio-video generation. This work proposes a multi-scale dual-stream spatio-temporal autoencoder to encode both video and audio into a unified latent space. The 2D latents from both modalities are stacked together and processed by a spatio-temporal diffusion transformer to generate the final output. The results on the Landscape and AIST++ benchmarks show its impressive performance across various evaluation metrics.

**Strengths:**

1. Design a preprocessing method to convert audio into video-like-spectrogram to mitigate its structural inconsistency with video.
2. Propose a multi-scale dual-stream spatio-temporal autoencoder to perform an orthogonal projection of 3D latent data into three separate 2D latents, creating a more efficient and expressive audio and video latent representation.
3. This work provides compelling empirical evidence of model superiority. It outperforms previous methods on all metrics like FVD, KVD, FAD.

**Weaknesses:**

1. Lack of evaluation metrics: audio-video semantic and temporal alignment should be evaluated, such as ImageBind-AV, AV-Align, DeSync and so on. In addition, the text-audio and text-video consistency score should be also evaluated.
2. The paper claims its method avoids reliance on a neural vocoder by using a direct "inverse Mel transformation". This is an important efficiency claim, but it lacks detail.
3. Did you compare with separate audio and video VAEs setting?
4. The inference efficiency is just benefited by reduced inference step? Is there any specific design for speeding up the inference?
5. This work only considers the label-condition generation and ignore the text as the control.
6. Some details of human evaulation are missing: How many volunteers? What was the expertise level of the annotators? What's defination for AV alignment, semantic or temporal alignment?
7. The demo and implementation code are not provided.

**Questions:**

1. Missing alignment evaluation metrics.
2. Unclear description about avoiding reliance on a neural vocoder by using a direct "inverse Mel transformation"
3. Missing some comparison experiments, such as separate VAEs setting and text-condition.
4. The improvement of inference efficiency is unclear.
5. The details of human evaluation are missing.

---

### Official Review · Reviewer_jMCo · 2025-10-31

**Soundness:** 1
**Presentation:** 3
**Contribution:** 2
**Rating:** 2
**Confidence:** 5

**Summary:**

This paper tackles the task of Sounding Video Generation (SVG), the simultaneous synthesis of video and its synchronized audio. The authors identify the structural heterogeneity between modalities, the difficulty of achieving fine-grained spatiotemporal alignment, and high computational costs as key challenges.

To address these, the paper proposes PAV-DiT, a novel framework based on a diffusion transformer operating in a latent space. The core of the proposed method is a Multi-scale Dual-stream Spatio-temporal Autoencoder (MDSA), which aims to bridge audio and video into a unified cross-modal latent space. The framework first transforms raw audio into a "video-like" representation (stacked spectrograms) and then uses MDSA with an orthogonal decomposition mechanism to project both modalities into a shared set of 2D latents. A Spatio-Temporal Diffusion Transformer (STDiT) then acts as the generator within this latent space.

The authors claim that this approach improves cross-modal fusion and efficiency. Based on experiments on the Landscape and AIST++ datasets, they report state-of-the-art results on all evaluation metrics, surpassing prior methods in both generation quality and computational speed.

**Strengths:**

*   **Addresses a Challenging and Important Problem**: The paper focuses on the task of synchronized audio-video generation, which is a challenging, important, and highly relevant problem in multimodal content creation. The authors correctly identify key difficulties in this domain, such as cross-modal alignment and computational efficiency, providing a clear motivation for their work.

*   **Novelty in Architectural Design**: The paper proposes a novel and complex architecture, particularly in the design of the Multi-scale Dual-stream Spatio-temporal Autoencoder (MDSA). The concept of using orthogonal projection to decompose 3D spatiotemporal data into a set of 2D latent representations is an interesting and non-trivial architectural idea aimed at improving efficiency and disentangling modality features.

*   **Strong Reported Performance on Benchmarks**: According to the quantitative results presented in Table 1, the proposed PAV-DiT model achieves state-of-the-art performance on the Landscape and AIST++ benchmarks. The reported improvements on key metrics like FVD and FAD are substantial, and the claimed gains in inference speed are also significant, suggesting the potential effectiveness of the proposed approach.

**Weaknesses:**

While the paper presents a novel architecture with clear motivation, several aspects concerning its methodological clarity and experimental validation prevent a confident assessment of its contributions.

1.  **Insufficient Experimental Validation for Key Claims**: The paper makes several strong claims about the benefits of its design, but these are not adequately supported by targeted experiments or ablation studies.
    *   **Unverified Claims on Alignment and Fusion**: The contributions section claims the method improves "cross-modal alignment and temporal coherence" (line 105), "reduces redundancy" for "efficient fusion" (line 106-107), and enhances "motion consistency and synchronization" (line 110). However, the paper does not provide specific experiments to validate these claims. For instance, how is the improvement in temporal coherence measured beyond the final FVD/FAD scores? How is the reduction in redundancy quantified? Without direct evidence, these claimed benefits remain speculative.
    *   **Lack of Validation for the MDSA Autoencoder**: The MDSA is a core contribution, yet its effectiveness is not properly isolated and validated. A crucial missing experiment is a quantitative and qualitative comparison of its reconstruction capabilities against standard or separate VAE/VAE models for video and audio.

2.  **Narrow and Incomplete Experimental Comparison**: The paper exclusively compares PAV-DiT against other joint-generation models. While the introduction dismisses "cascade generation" methods due to potential misalignment issues (lines 039-043), this claim is not empirically verified. A more comprehensive evaluation would necessitate including a strong cascade baseline (e.g., a video model like Wan 2.1 followed by a video-to-audio model like MMAudio or AudioX) to provide a direct comparison against the cascade generation paradigm.

3.  **Lack of Clarity in Methodology and Terminology**: The paper's description of its methodology contains ambiguities that hinder full understanding. For example, key notations in the architectural diagrams (e.g., "HT, WT") are not defined, and the recurring use of the imprecise term "video-like audio representations" for what appears to be a standard sequence of mel-spectrograms could be confusing.

4.  **Absence of Qualitative Samples**: For a task where perceptual quality is critical, the submission lacks supplementary video samples. This omission makes it impossible for reviewers to subjectively verify the claimed improvements in audio-visual synchronization, quality, and motion consistency, which are not fully captured by objective metrics alone.

**Questions:**

1.  **On the Cascade Generation Baseline**: The paper argues that cascade-based approaches are inferior but does not provide a direct comparison. To substantiate this claim, could authors provide experimental results comparing PAV-DiT against a strong cascade baseline?

2.  **On the Justification of MDSA**: The MDSA is presented as a core contribution. To better isolate its benefits, could you provide a quantitative comparison of its reconstruction performance against standard autoencoders for video and audio respectively?

3.  **On Substantiating Design Claims**: How were the specific claims of improving "temporal coherence," "reducing redundancy," and enhancing "motion consistency" validated beyond the final FVD/FAD scores? Are there specific ablation studies or metrics that demonstrate these points directly?

The authors address an important and challenging problem in synchronized audio-video generation (SVG). The motivation for a unified, efficient framework is well-founded, and the architectural ideas within the MDSA show considerable effort.

However, as detailed in my review, the paper currently suffers from significant weaknesses, primarily centered around the lack of rigorous and comprehensive experimental validation. The claims of superiority against cascade-based methods, as well as claims on specific aspects like "redundancy reduction" and "temporal coherence", are not sufficiently backed by targeted experiments, ablations, or standard metrics. Furthermore, the lack of clarity in key parts of the methodology and the absence of qualitative samples make it very difficult to confidently assess the true performance and contribution of the proposed system.

Therefore, my initial recommendation is **Reject**.

---

### Official Review · Reviewer_ABui · 2025-11-01

**Soundness:** 2
**Presentation:** 2
**Contribution:** 1
**Rating:** 2
**Confidence:** 3

**Summary:**

This paper proposes PAV-DiT, a unified diffusion transformer framework for sounding video generation (SVG) that jointly synthesizes synchronized video and audio. The method introduces a Multi-scale Dual-stream Spatio-temporal Autoencoder (MDSA) that encodes both modalities into projected 2D latent spaces through orthogonal decomposition, followed by a Spatio-Temporal Diffusion Transformer (STDiT) for generation in the joint latent space. While the paper is technically ambitious and draws on strong recent trends (latent diffusion, cross-modal transformers, flow sampling), the contribution is largely incremental to existing multimodal diffusion literature, and the writing overstates novelty.

**Strengths:**

- The proposed autoencoder achieves visually coherent reconstructions, showing solid engineering effort.
- The inclusion of human evaluation provides minimal qualitative validation, though limited in scale.

**Weaknesses:**

- The architecture combines existing ideas (latent diffusion + multi-scale attention + DiT) without introducing fundamentally new modeling principles.
- The paper provides no statistical tests, missing open-domain synchronization metrics, and limited audio evaluation (e.g., no PESQ, STOI, or MOS).
- “State-of-the-art” is asserted without clear reproducibility or code release.
- Figures are cluttered, descriptions overly mechanical, and English awkward in places.
- The authors miss theoretical insight into why orthogonal decomposition aids alignment or how multi-scale attention specifically benefits synchronization.

**Questions:**

- How does orthogonal projection (Eq. 2–3) compare empirically to simpler shared-latent autoencoders like MM-LDM?
- Could the authors report quantitative synchronization metrics (e.g., audio–visual alignment error, temporal offset correlation)?
- How does the model generalize to unseen domains or longer videos beyond 16 frames?
- Is the Reflected Flow Sampling integrated into training or only used for inference acceleration?
- Can the authors release generated audio samples for qualitative verification of temporal coherence?

---

### Official Review · Reviewer_v47B · 2025-11-01

**Soundness:** 2
**Presentation:** 2
**Contribution:** 2
**Rating:** 4
**Confidence:** 4

**Summary:**

The paper proposes PAV-DiT, a diffusion transformer for audio-video generation. It starts by converting audio into a video-like representation: the waveform is split per video frame, made into a Mel-spectrogram, and stacked to match the video's T×H×W structure. A Multi-scale Dual-stream Spatio-temporal Autoencoder (MDSA) encodes audio and video into three 2D latents aligned with the temporal, height, and width axes. This axis-wise projection is inspired by tensor factorization. It achieves FVD/KVD/FAD of 80.3/7.3/8.5 on Landscape and 77.6/18.2/9.4 on AIST++. A human study on 1,500 samples with two raters shows higher AQ, VQ, and A‑V.

**Strengths:**

* The video‑like audio construction removes modality shape mismatch, enabling symmetric encoders and simpler cross‑modal fusion.
* Projecting 3D features onto $(t,h,w)$-aligned 2D latents is an elegant way to reduce token counts while encouraging separable modeling.
* MT‑SelfAttn (×1/×2/×4) plus GCM‑Attn and Bi‑Block CrossAttn improve synchronization; ablations show each component lowers rFVD/rKVD.
* Standard benchmarks, an open‑domain AudioSet‑100K test, qualitative comps (Fig. 3–4, 8–9), and a user study provide a rounded evaluation.

**Weaknesses:**

* The abstract claims surpassing prior work “across all metrics,” yet AV‑DiT reports FVD = 68.8 on AIST++, better than PAV‑DiT’s 77.6 at 256² (Table 1, p. 7). The global SOTA claim is invalid.
* All results are 16 frames at 256²; there’s no study of longer clips, higher resolutions, or variable FPS. AudioSet‑100K selection/splits and leakage checks are only briefly mentioned.
* There are two annotators mentioned, but no inter‑rater agreement is provided.
* No video demo provided.

**Questions:**

* How do you measure independence among $z_t, z_h, z_w$? Would decorrelation losses help validate the decomposition (Eqs. 1–3)?
* What happens beyond 16×256²—e.g., >64 frames, 512², or variable FPS? Is the six‑frame stacking still optimal, or does the axis tokenization need to change.
* What's the inter‑rater agreement between annotators?
* Why didn't author provide any video demo?

---

### Note · Authors · 2025-11-14

I have read and agree with the venue's withdrawal policy on behalf of myself and my co-authors.